# Validation of the Eating Disorder Examination Questionnaire in Danish Eating Disorder Patients and Athletes

**DOI:** 10.3390/jcm10173976

**Published:** 2021-09-02

**Authors:** Mia Beck Lichtenstein, Lauge Haastrup, Karen Krogh Johansen, Jacob B. Bindzus, Pia Veldt Larsen, René Klinkby Støving, Loa Clausen, Jakob Linnet

**Affiliations:** 1Department of Clinical Research, University of Southern Denmark, 5000 Odense C, Denmark; 2Centre for Telepsychiatry, Mental Health Services in the Region of Southern Denmark, 5000 Odense C, Denmark; laugehaastrup@gmail.com (L.H.); Karen.krogh.johansen@rsyd.dk (K.K.J.); jacob.bindzus@outlook.dk (J.B.B.); Jakob.Linnet@rsyd.dk (J.L.); 3Mental Health Services in the Region of Southern Denmark, 5000 Odense C, Denmark; pia.veldt.larsen@rsyd.dk; 4Center for Eating Disorders, Odense University Hospital, 5000 Odense C, Denmark; rene.stoeving@rsyd.dk; 5Department of Child and Adolescent Psychiatry, Aarhus University Hospital, 8200 Aarhus N, Denmark; loaclaus@rm.dk; 6Department of Clinical Medicine, Aarhus University, 8200 Aarhus N, Denmark

**Keywords:** Eating Disorder Examination Questionnaire, eating disorders, validation, factor structure, exercise, sport

## Abstract

The Eating Disorder Examination Questionnaire (EDE-Q) is a gold standard questionnaire to identify eating disorder symptoms but has not yet been validated in Danish. The scale consists of four theoretical constructs of disordered eating: Restraint eating, Eating concerns, Shape concerns and Weight concerns. However, the four-factor structure has been difficult to replicate across cultures. This study aimed to examine the factor structure and psychometric properties of the EDE-Q in Danish. The study consisted of four samples (aged 15–70): Patients with anorexia, bulimia and unspecified eating disorders (*n* = 101), patients with symptoms of binge-eating disorder (*n* = 300), recreational athletes (*n* = 404), and elite athletes (*n* = 526). Depending on the analysis performed, participants had to complete the EDE-Q, the SCOFF questionnaire for eating disorders or the Binge Eating Disorders Questionnaire. In accordance with international research, we found no evidence for a four-factor structure in the EDE-Q among patients or among athletes. But our results showed significant, positive associations between EDE-Q and SCOFF, BED-Q and MDI in all samples. We conclude that the internal structure of EDE-Q is low, while construct validity is high, making EDE-Q useful as an instrument to identify individuals with eating disorder symptoms, including recreational, and elite athletes.

## 1. Introduction

The Eating Disorder Examination Questionnaire (EDE-Q), originally developed by Fairburn and Beglin [1], measures behavioral and cognitive symptoms of eating disorders, including binge eating, self-induced vomiting, excessive exercise and negative body image [2]. 

The questionnaire is based on the Eating Disorder Examination Interview (EDE-I), which is the gold standard in eating-disorder assessment. The EDE questionnaire (EDE-Q) was developed to provide a straightforward and quick assessment tool and is one of the most used instruments to screen for symptoms of eating disorders [3]. It consists of four subscales measuring 1. Restraint, 2. Eating concern, 3. Shape concern, and 4. Weight concern. The EDE-Q has a global score that is an average of the four subscales and is used as a cut-off point for eating-disorder pathology.

The scale has been validated across several different languages. Berg et al. found that the EDE-Q had acceptable psychometric properties in a literature review of English-speaking adult samples from different countries [2]. Other studies have examined the reliability or validity of the EDE-Q in Spanish [4,5,6], Japanese [7], Turkish [8], Persian [9], Finnish [10], Hebrew [11], Greek [12], Norwegian [13,14], Swedish [15], French [16] and German [17]. However, the EDE-Q has never been validated in Danish. A Danish validation of the EDE-I was conducted by Kjeldberg et al. [18] based on 1586 participants from a clinical eating-disorder sample (all female). An exploratory factor analysis was conducted and suggested both one-, two- and three-factor models, depending on the sub-sample. The factor of Food Restraint seemed consistent among the groups. 

The original theoretical four-factor structure of EDE-Q has been questioned in several empirical studies. A review by Berg et al. found three studies evaluating the factor structure of the EDE-Q, none supporting the original four-factor model [2]. One of the studies compared the original four-factor model to a three-, two- and one-factor model and found the one-factor solution to be the best fit. Another study found a three-factor solution consisting of 1. Dietary Restraint, 2. Shape/Weight Over-evaluation and 3. Body Dissatisfaction. The third study found support for a two-factor model, consisting of a Restraint factor and a combined Eating–Shape–Weight concern factor. 

The EDE-Q factor structure has been examined empirically since the review by Berg et al., but the studies have not been reviewed and synthesized [2]. Furthermore, the factor structure of EDE-Q has not been evaluated in Danish yet. 

Most studies report a higher prevalence proportion of eating disorder symptoms among athletes compared to the general population [19,20]. While several studies have used the EDE-Q to assess the frequency of eating disorders in athletes, only one study has validated the EDE-Q in sport [21]. The authors concluded that a three-factor solution provided the best fit for all groups, except for the sedentary males. They also emphasized that athletes may display the same behaviors towards weight and shape as sedentary people but differ according to diet and eating concerns. More research investigating the validity of the EDE-Q in athletic populations is requested.

The main aim of this study was to conduct a confirmatory factor analysis of the EDE-Q in a Danish sample of patients with eating disorders and a sample of recreational and elite athletes. Secondly, we aimed to assess the construct validity of EDE-Q by comparing the EDE-Q global score with other measures of eating pathology and depression. Based on existing research we expected the factor structure of EDE-Q to be inconsistent, but we assumed the instrument had a modest-to-high positive correlation with other measures of eating disorders and depression.

Before conducting the psychometric analyses we reviewed the literature testing factor models of the Eating Disorder Examination Questionnaire (EDE-Q).

## 2. Materials and Methods

The review study was conducted as a literature review (or rapid review) using the following search to create a brief overview of the findings of factor models: EDE-Q AND Athlete*s OR sport athlete*s AND eating disorder* OR anorexia nervosa OR bulimia nervosa OR binge eating disorder OR unspecified eating disorder AND factor analysis OR factor exploration. We only included English language publications in the period 2012–2021.

The empirical research design was a questionnaire-based cross-sectional data-collection consisting of four sub-samples:Individuals with Anorexia Nervosa (AN), Bulimia Nervosa (BN), and unspecified eating disorders;Individuals seeking treatment for mild-to-moderate Binge Eating Disorder (BED) in a Mental Health Department;Elite athletes competing at the highest national and international level;Recreational athletes practicing physical activity on a weekly basis.

### 2.1. Participants

The first subsample consisted of 101 individuals with self-reported AN (*n* = 55), BN (*n* = 10), and unspecified eating disorders (*n* = 36) recruited from national patient organizations, communal health services or patients admitted to psychiatric wards in Denmark. Recruitment occurred during April 2020 and November 2020.

The second subsample included 300 participants seeking a public internet-based treatment program for mild to moderate BED. Participants were recruited between May 2019 and November 2020.

The third subsample consisted of 404 Danish elite athletes across 15 different sports; the most common were cycling, track and field, handball, football, karate, swimming, triathlon and gymnastics. The participants completed the questionnaire between December 2019 and March 2020 and were part of a large cross-sectional study on mental health in Danish elite athletes [22]. 

The fourth subsample consisted of 526 Danish recreational athletes recruited through social media channels between March and May 2020. A description of the study including a link to the survey was posted on various exercise groups on Facebook. The most common sports were: running (recreational), fitness (power and endurance), track and field, handball, triathlon, cycling, gymnastics and boxing.

### 2.2. Measures

All participants completed the EDE-Q and the Major Depressive Inventory (MDI) in Danish. The sample of eating-disorder patients, recreational athletes, and elite athletes also completed the eating-disorder screening instrument, SCOFF. The BED sample completed the Binge Eating Disorder Questionnaire (BED-Q) developed in-house to assess BED symptoms. See Appendix A.

We used a Danish version of the EDE-Q, translated by Elsass et al. [23] with back translation approved by Christopher Fairburn. The EDE-Q has 28 items and four subscales measuring 1. Restraint, 2. Eating concern, 3. Shape concern and 4. Weight concern. A total score called “Global Score” is an average of the subscales. 

SCOFF is a five-item screening tool designed to assess basic symptoms of BN and AN [24]. The measure is scored on a dichotomous “yes” or “no” scale and two or more positive answers indicate risk of an eating disorder. A meta-analysis of the diagnostic accuracy of the SCOFF scale showed excellent performance [25]. The SCOFF has been translated and validated in Danish showing reasonable sensibility and sensitivity [26].

The Binge Eating Disorder Questionnaire (BED-Q) was developed in-house to assess symptoms of BED. It is a 9-item scale based on the diagnostic criteria for BED in the Diagnostic and Statistical Manual of Mental Disorders (DSM-5) [27]. The BED-Q is attached as an appendix.

The MDI contains 10 items that cover symptoms of depression defined by the International Statistical Classification of Diseases and Related Health Problems 10th edition (ICD-10) [28]. Answers are given on a 6-point Likert scale, and the total score ranges from 0 to 50. Recommended cut-off points are 21 for mild depression, 26 for moderate depression, and 31 for severe depression [29]. The scale has been validated and shows good psychometric properties in several studies including Danish [29,30].

### 2.3. Ethics

By completing the questionnaire, participants (or parents for participants aged 15–17 years) gave their informed consent for scientific use. No personal data, such as names, birthdays, or e-mail addresses were mandatory to complete the questionnaire. Data collection based on anonymous questionnaires does not need ethical approval, as confirmed by The Regional Committees on Health Research Ethics for Southern Denmark (approval number 20212000-57 and 20192000-145).

### 2.4. Statistical Analyses

Analyses were performed in SPSS version 26 and STATA 16.1 (Statacorp, College Station, TX, USA). Descriptive statistics on participant characteristics were computed. Chi square analyses were used for dichotomous variables and *t*-tests were used for comparing mean scores.

The factor model comprising the four EDE-Q factors (Restraint (EDE-Q Items 1–5), Eating concern (Items 7, 9, 19, 20, 21), Shape concern (Items 6, 10, 11, 23, 26, 27, 28, Note: not including Item 8) and Weight concern (Items 8, 12, 22, 24, 25)) was analyzed in two combined samples: one combining the eating disorder and BED samples, and one combining the elite and recreational samples. Structural equation models (SEM) were used to conduct confirmatory factor analyses (CFA) with standard errors estimated using non-parametric bootstrap sampling with 1000 replications, to account for non-normality of the latent factors. Sensitivity analyses were conducted of the CFAs with Item 8 attributed to the factor on Shape concern rather than Weight concern.

The goodness of fit of the factor models was assessed by means of comparative fit index (CFI), Tucker–Lewis index (TLI), root-mean-square error of approximation (RMSEA), and standardized root-mean-square residual (SRMR). Further, Cronbach’s α was used to assess the internal consistency of each of the four factors. 

Construct validity was assessed by using Pearson and Spearman correlations between EDE-Q and other measures of psychopathology SCOFF/BED-Q and MDI.

## 3. Results

### 3.1. Literature Review

Our literature review resulted in 19 publications from 2012 to 2021 conducting a factor analysis of the EDE-Q. The studies are presented in Table 1 and show remarkable inconsistency across samples. There is no evident factor structure, and the studies display both one-, two- and three-factor models.

Different shorter versions (e.g., seven items and eight items) have been created and suggested, but none have been replicated sufficiently to conclude and recommend new evident factor solutions.

### 3.2. Internal Consistency

The descriptive characteristics of participants in the empirical studies are presented in Table 2 according to age, gender, Body Mass Index (BMI) and clinical scores.

The Cronbach’s alpha coefficient was low, but acceptable, in the eating disorder sample ranging from 0.61 (weight concern scale) to 0.80 (restraint scale). In the combined sport sample, the alpha coefficient ranged from 0.81 (restraint scale) to 0.91 (shape concern scale). See Table 3.

The results of the CFA showed no optimal fit of the original four-factor model in the two pooled samples, as the only acceptable test was SRMR in the athlete sample. Accepting correlations between items 7 and 8 and items 22 and 23 that are specifically closely related did not change the results significantly, despite showing a better fit. 

Sensitivity analyses with item 8 attributed to the shape concern factor resulted in very similar results.

### 3.3. Construct Validity

When examining the construct validity of EDE-Q by comparing the relationship between EDE-Q global scale and SCOFF total score (or BED-Q for the BED sample), the results demonstrated positive and significant correlations in all samples. Further, we examined the relationship between EDE-Q and MDI, and found positive and significant correlations in all samples. The results and level of significance are shown in Table 4.

## 4. Discussion

### 4.1. Literature Review and Factor Structure of EDE-Q

In accordance with other validation studies our Danish study could not empirically support the original factor structure (restraint eating, eating concern, shape concern and weight concern) presented as a theoretical model by Fairburn [1]. 

The results from the EDE-Q literature across several countries (Table 1) indicate that no evident and consistent factor structure of the original EDE-Q can be identified. Models suggesting one-factor, two-factor and three-factor structures have been presented and evaluated. Shorter versions of EDE-Q have been developed, but the internal structure of the scale differs across studies and samples. Several studies point at a seven-item scale, but also 8 items, 15 items (for female), 16 items (for male), 18 items and 22 items have been proposed during the last 10 years.

The scale was not originally developed from empirical research, but is a theoretical construct that is reasonable from a clinical perspective. The confusion of results may be related to the interpretation of constructs by different populations. Restraint eating can be part of a healthy diet in general populations or in athletic settings, while it is pathological in eating-disorder patients. Weight concerns may be relevant in severely overweight BED patients, while it they are related to disturbed body image in AN patients. Eating concerns may appear differently in clinical eating disorder samples compared to athletic samples. Darcy et al. also suggested that athletes may differ in diet and eating concerns compared to sedentary people [21].

Furthermore, a large number of the elite athletes in the current study reported a 0-score on several items, leading to a high Cronbach’s Alpha coefficient but a weak factor model. The wording of weight and shape concerns may overlap and can be difficult to separate, leading to similar scores that confuse the CFA model, because some items (e.g., 7 and 8) are loading on two factors.

Despite the lack of psychometric evidence, the four constructs might be useful in a clinical setting if patients find them coherent. The face validity of EDE-Q and EDE-I may be high in the assessment of pathological thoughts and behaviors in collaborations between clinicians and patients. For athletes, the EDE-Q may increase the awareness of disturbed eating patterns and body image. Qualitative research investigating the interpretation of items and factors is recommended in future research.

Further studies of the factor structure may confuse the landscape of psychometric reliability. Instead, we should evaluate the validity of EDE-Q because the instrument may be useful in identifying eating disorders. 

### 4.2. Validity of the EDE-Q

Despite lack of internal consistency, the construct validity was confirmed in all samples, as EDE-Q scores were highly correlated with other eating-disorder measures and depressive symptoms. This is in accordance with the American and Hebrew validations of the EDE-Q demonstrating positive and significant correlations between EDE-Q and other health measures, including depressive symptoms [33] and eating disorder symptoms [11]. The German 8-item scale also showed good construct validity with significant positive correlations between EDE-Q and the Eating Attitudes Test [17].

In conclusion, the studies support the ability of the EDE-Q to cover the construct of eating disorders, making it a valid tool for identifying eating disorders, though the lack of evidence of the four-factor model should be considered.

### 4.3. Limitations

It may be a limitation that the four sub-samples in our study differed according to sample size, age, gender, BMI and recruitment method. 

Limitations must be taken as the study comprise heterogeneous samples of patients and amateur exercisers which might have affected the results. On the other hand, the factor structure ideally ought to be the same if the EDE-Q should be suggested as a solid questionnaire. 

We have chosen a wide range of ED’s from AN with very low BMI to BED with very high BMI. Although, this per see increases the external validity of the work, few extreme patients potentially may distort the statistics.

The EDE-Q has been questioned in male samples, and there is a risk of measurement bias as the scale may not comprehensively assess all domains of disordered eating relevant to males [45].

The study might have benefited from adding a fifth group of community controls. However, the EDE-Q was developed to assess symptoms in a clinical setting and risk groups and testing the factor structure in these groups was considered first step.

Despite these concerns, the psychometric properties of the scale were similar across sub-samples, indicating that it is useful across diagnoses and in sport settings.

An exploratory analysis or another CFA model could have accounted for the inability of our CFA to provide an optimal fit of the original four-factor structure. However, the aim of our study was only to validate the existing gold standard scale in Danish and not to explore if another structure could be supported.

It should be mentioned as a potential bias that some of our data were collected during the COVID-19 pandemic. A retrospective study from Italy [46] found that ED symptoms increased in ED patients during March and April 2020. However, the crisis was worse in Italy than in Denmark, including more restrictions and more impact on hospitals, public health and mortality.

## 5. Conclusions

The psychometric evaluation of the EDE-Q in a Danish sample of eating-disorder patients and a sample of athletes did not find evidence for the original four-factor structure. Our literature review further depicted the variance regarding number of factors (one, two or three) and items suggested (from 7 to 22) in previous validation studies. Thus, the four-factor internal structure of EDE-Q may be understood as a theoretical model useful in clinical settings, but it is not supported by empirical data.

However, the construct validity was high in all samples, indicating that the scale has the potential to identify individuals with eating pathology and comorbid mental distress. Development of a sport-specific tool may present with higher reliability and validity. Until then, the EDE-Q is a usable screening tool, but with caution and awareness of its limitations.

## Figures and Tables

**Table 1 jcm-10-03976-t001:** Factor structure of the Eating Disorder Examination Questionnaire in cross-cultural samples in the period 2012–2021.

Authors	Year	Country	Sample	Factor Solution and Item Number
Barnes et al. [31]	2012	UK	569 adult participants including 403 university students (91.8% female) and 166 eating disorder patients (95.8% female). Age is not mentioned.	A three-factor model suggested by Peterson et al. 2007 found best fit for both groups: (1) Weight and shape concern, (2) Eating concern and (3) Restraint.
Penelo et al. [32]	2013	Mexico	2928 schoolchildren (1544 females, 1384 males), mean age 15.2 (SD = 1.79) and 15 (SD = 1.78) years for females and males, respectively.	A two-factor model was suggested as the best fit: (1) A combination of eating weight shape concerns and (2) Restraint.
Giovazolias et al. [12]	2013	Greece	664 participants including 500 university students (all female), mean age 20.55 (SD = 3.27) years, and 164 psychology students (all female), mean age 20.90 (SD = 3.29) years.	Three-factor models were found to have the best fit: (1) Combined Shape- and Weight Concern factor, (2) Restraint factor, and (3) Eating Concern factor.
Darcy et al. [21]	2013	USA	1637 university students separated into two groups: 976 competitive athletes (544 female, 432 male) and 858 non-athletes (429 female, 229 male). Mean age 20.87 (SD = 1.66) years.	A three-factor model provided the best fit, except for non-athlete male participants where the best fit was a two-factor model.
Grilo et al. [33]	2014	USA	801 University students (573 female, 228 male), mean age 20 (SD = 2.5) years.	Evidens for a seven-item, three-factor model: (1) Dietary restraint, (2) Shape/weight overvaluation and (3) Body dissatisfaction.
White et al. [34]	2014	UK	Two adolescent samples: (1) 458 participants (257 female, 201 male), mean age 15.3 (SD = 1.18) years, and (2) 259 participants (265 female, 194 male), mean age 15.2 (SD = 1.18) years.	A three factor, 22-item model was found as best fit: (1) Shape and weight concerns, (2) Restriction and (3) Preoccupation and eating concern.
Carrard et al. [16]	2015	France	116 patients with BED (all female), mean age 38.5 (SD = 11.4), and 161 participants without a eating disorder (all female), mean age 28.1 (SD = 8.1) years.	A seven-item, three-factor model suggested by Grilo et al. 2013 [33] revealed an adequate fit.
Kliem et al. [17]	2016	Germany	2520 participants from 2009 study (1334 female, 1174 male), mean age 49.67 (SD = 18.30) and 2508 participants from 2013 (1354 female, 1166 male), mean age 49.90–51.02 (SD = 18.20–18.66) from the general population.	An eight-item with second-order corrected factors found the best fit.
Forsén Mantilla et al. [15]	2017	Sweden	487 school children (239 female, 248 male), mean age ≈13.50 (SD = 0.50) years, and 570 patients with eating disorders (all female), mean age 13.46 (SD = 0.69) years.	One underlying factor was found for adolescent girls: Dissatisfaction with shape and weight. For boys three factors were found: (1) Weight-related concerns, (2) Body discomfort, and (3) Restraint.
Zohar et al. [11]	2017	Israel	292 community volunteers (241 female, 51 male), mean age = 33.39 (SD = 14.52).	A three-factor model gave the best fit with weight and shape concerns converted into a single factor.
Machado et al. [35]	2018	Portugal	Two samples: (1) 2026 high school students (all female), mean age 16.2 (SD = 1.34) years, and 2091 college students (all female), mean age 23.8 (SD = 9.16) years, (2) 609 clinical participants seeking treatment for an eating disorder (592 female, 17 male), mean age 23.8 (SD = 9.16) years.	A three-factor, seven-item model suggested by Grilo et al., 2013 [33] was found to have the best fit for both groups.
Serier et al. [36]	2018	USA	561 university students (all female), 336 Hispanic and 225 non-Hispanic, mean age = 20.11 (SD = 3.43).	A three-factor seven-item structure suggested by Grilo et al., 2013 [33] was found to have the best fit for both groups.
Carey at al. [37]	2019	UK	2459 participants from three samples: (1) 1075 students (851 female, 224 male), mean age 19.89 (SD = 1.98), (2) 653 students (489 females, 164 males), mean age 22.16 (SD = 3.69) and 22.86 (SD = 3.69) years respectively, and (3) 731 non-students (561 female, 179 male), mean age 32.68 (SD = 10.25) and 34.39 (SD = 11.08) years respectively.	A three-factor model had the best fit for both gender using an 18-item for female and a 16-item model for male participants: (1) Shape and weight concern, (2) Preoccupation and eating concern and (3) Restriction.
Compte et al. [38]	2019	Argentina	Four samples: (1) 232 college students (all male), mean age 23.53 (SD = 5.61) years, (2) 286 weightlifters (all male), mean age 29.24 (SD = 9.27) years, 3) 279 cross-fit gym users (all male), mean age 29.86 (SD = 7.41) years, and (4) 203 rugby players (all male), mean age 21.71 (SD = 3.45) years.	A one-factor, eight-item model was found to have the best fit between the groups.
Sepúlveda et al. [39]	2019	Spain	167 adolescent patients with an eating disorder (all female), mean age 15.45 (SD = 1.59) years.	The two-factor structure suggested by Penelo et al., 2013 [32] was found to have the best fit.
Lewis-Smith et al. [40]	2020	India	1413 adolescents (635 females, 778 males) from an urban private school, mean age 13 (SD = 0.84) years.	A two-factor model, 15 items for girls and 18 for boys, had the best fit: (1) Preoccupation and control, and (2) Weight and shape concerns
Rand-Giovanetti et al. [41]	2020	Australia	940 psychology students (69.9% female, 29.6% male, 1% other), mean age 20.34 (SD = 3.74) years.	The four-factor 22-item structure suggested by Friborg et al. 2015 was found to have the best fit: (1) Dietary restraint, (2) Preoccupation and restriction, (3) Weight and shape concern, and (4) Eating shame.
Scharmer et al. [42]	2020	USA	703 participants from the general population (all male), mean age 33.76 (R = 18–67) years. 70.2% identified as heterosexual while 29.8% identified as a sexual minority (homosexual, bisexual, or queer).	The three-factor seven-item structure suggested by Grilo et al. (2013) [33] was found to have the best fit for both groups.
Klimek et al. [43]	2021	USA	962 cisgender individuals from a sexual minority (483 female, 479 male), mean age 23.68 (SD = 3.73) years.	Two different model were found to have an adequate fit: A three-factor, seven-item structure suggested by Grilo et al. (2013) [33] and a four-factor, 22-item structure suggested by Friborg et al., 2015.
Rica et al. [44]	2021	Spain	850 university students (all male), mean age 19.8 (SD = 2.8) years.	A two-factor had the best fit: (1) Restraint, and (2) Weight and shape eating concern.

SD = standard deviation; R = range from youngest to oldest age of participants; BED = Binge Eating Disorder.

**Table 2 jcm-10-03976-t002:** Demographic characteristics, eating-disorder symptoms and depressive symptoms.

		Eating Disorder Patients (*n* = 101)	Binge Eating Disorder (*n* = 300)	Elite Athletes (*n* = 404)	Amateur Exercisers (*n* = 526)
Age range years		15–64	18–68	15–47	15–70
Age mean		27.6 (SD 9.9)	39.1 (SD 11.5)	20.0 (SD 4.7)	29.6 (SD 10.4)
Gender female/male		97/4	271/29	207/197	364/162
Body Mass Index		22.4 (range 7.6–54.3)	37.8 (range 16.6–76.2)	21.8 (range 16.6–30.8)	23.7 (range 15.2–42.0)
EDE-Q					
	Restraint eating	3.62 (SD 1.84)	2.63 (SD 1.30)	0.83 (SD 1.16)	1.28 (SD 1.41)
	Eating concerns	3.23 (SD 1.56)	3.56 (SD 1.10)	0.46 (SD 0.89)	0.76 (SD 1.12)
	Shape Concerns	4.57 (SD 1.40)	4.60 (SD 0.92)	1.20 (SD 1.28)	1.59 (SD 0.16)
	Weight concerns	4.25 (SD 1.49)	4.37 (SD 0.91)	1.08 8SD 1.02)	1.46 (SD 0.06)
	Global score	3.92 (SD 1.42)	3.79 (SD 0.83)	0.89 (SD 1.02)	1.25 (SD 0.05)
SCOFF 2+		95.9%	NO	17.1%	31%
BED-Q total		NO	17.17 (SD 3.68)	NO	NO
Depression mean score		29.7 (SD 10.6)	24.8 (SD 9.1)	8.97 (SD 7.4)	12.9 (SD 9.7)
Mild depression		13.5%	16.3%	4.0%	6.8%
Moderate depression		11.5%	19.0%	2.0%	4.4%
Severe depression		52.1%	30.7%	2.7%	7.8%

SD = Standard Deviation; NO = Not Available. EDE-Q = Eating Disorder Examination Questionnaire. BED-Q = Binge Eating Disorder Questionnaire.

**Table 3 jcm-10-03976-t003:** Summary of goodness-of-fit statistics.

	Cronbach’s α				Confirmatory Factor Analysis with No Correlated Uniqueness					Confirmatory Factor Analysis with Two Correlated Uniquenesses between Items 7 & 8 and Items 22 & 23				
Samples	Restraint	Eating Concern	Shape Concern	Weight Concern	χ^2^ (df)	CFI	TLI	RMSEA (90%-CI)	SRMR	χ^2^ (df)	CFI	TLI	RMSEA (90%-CI)	SRMR
Eating disorder & BED	0.779	0.677	0.768	0.608	1585 (203)	0.660	0.614	0.130 (0.125, 0.136)	0.121	1104 (201)	0.778	0.745	0.106 (0.100, 0.112)	0.118
Elite & exercise	0.805	0.813	0.910	0.818	2723 (203)	0.821	0.796	0.116 (0.112, 0.119)	0.067	1777 (201)	0.888	0.871	0.092 (0.088, 0.096)	0.064

BED = Binge Eating Disorder. CFI = Comparative Fit Index. TLI = Tucker–Lewis Index. RMSEA = Root-Mean-Square Error of Approximation. SRMR = Standardized Root-Mean-square Residual.

**Table 4 jcm-10-03976-t004:** Pearson and Spearman correlations between EDE-Q global score and other measures of eating pathology or depression.

	Correlation with EDE-Q Global Score in the Four Samples			
Measure	Mixed Eating Disorders	Binge Eating Disorder	Elite Athletes	Regular Exercisers
SCOFF	R = 0.599, *p* < 0.001	NA	R = 0.729, *p* < 0.001	R = 0.765, *p* < 0.001
	Rho = 0.536, *p* < 0.001		Rho = 0.611, *p* < 0.001	Rho = 0.730, *p* < 0.001
BED-Q	NA	R = 0.347, *p* < 0.001	NA	NA
		Rho = 0.335, *p* < 0.001		
MDI	R = 0.606, *p* < 0.001	R = 0.451, *p* < 0.001	R = 0.569, *p* < 0.001	R = 0.686, *p* < 0.001
	Rho = 0.574, *p* < 0.001	Rho = 0.454, *p* < 0.001	Rho = 0.499, *p* < 0.001	Rho = 0.595, *p* < 0.001

R = Pearson’s R; Rho = Spearman’s rho; NA = Not Available. MDI = Major Depression Inventory.

## Data Availability

The data presented in this study are available by request.

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
