# Peer review of "Validation of the Eating Disorder Examination Questionnaire in Danish Eating Disorder Patients and Athletes"

_jcm, 2021, doi:10.3390/jcm10173976_

Round 1
Reviewer 1 Report
This study examined the validation of the EDEQ in Danish eating disordered patients and athletes aiming to test the factor models of the EDEQ, to do a confirmatory factor analysis, and the construct validity in the various samples. This is a very interesting paper, with proper methodology, well written and important findings regarding the EDEQ – as the mostly used instrument in eating disorders.
Minor revisions are needed in order to be recommended for publication.
Minor comments:
- Abstract, line 2: You write ….. the EDE-Q, the SCOFF questionnaire for eating disorders or the Binge Eating Disorders Questionnaire……; This sentence is confusing regarding the “or” and not knowing what you described in detail in your paper later on. So I would suggest to change it accordingly (e.g.): “Depending on the analysis performed, participants had to complete the EDE-Q, the SCOFF questionnaire for eating disorders or the Binge Eating Disorders Questionnaire”.
- Participants: It would be helpful to get further information regarding the kind of sports activity of 1) the elite athletes (name at least some of the 15) and 2) the recreational athletes.
- Measures: It is of interest why you did not include a control group of non-eating disordered and non-sports-active people in you analysis.
- Measures: Further, why did you not include various cut-off points (e.g. >3 as a general cutoff or the male-specific of > 1.68 for males) regarding your point of construct validity?
- Discussion, line 254: You state in this section that you psychometric evaluation did not find evidence for the original four-factor structure of the EDE-Q. It would be interesting what you suggest as consequences of this finding regarding the construction of the EDEQ as an instrument that is mostly used in eating disorder research.
Author Response
We would like to thank the editor and the reviewers for taking our manuscript into consideration. We are very thankful for all relevant comments, and we have responded to each of them. Thanks again for the collaboration about this article.
Reviewer 1:
Minor revisions are needed in order to be recommended for publication.
- Abstract, line 2: You write ….. the EDE-Q, the SCOFF questionnaire for eating disorders or the Binge Eating Disorders Questionnaire……; This sentence is confusing regarding the “or” and not knowing what you described in detail in your paper later on. So I would suggest to change it accordingly (e.g.): “Depending on the analysis performed, participants had to complete the EDE-Q, the SCOFF questionnaire for eating disorders or the Binge Eating Disorders Questionnaire”.
Response 1: We have changed the sentence in the abstract according to the reviewer's suggestion: “Depending on the analysis performed, participants had to complete the EDE-Q, the SCOFF questionnaire for eating disorders or the Binge Eating Disorders Questionnaire”.
- Participants: It would be helpful to get further information regarding the kind of sports activity of 1) the elite athletes (name at least some of the 15) and 2) the recreational athletes.
Response 2: We have added information about the kind of sport: The third subsample consisted of 404 Danish elite athletes across 15 different sports; the most common were cycling, track and field, handball, football, karate, swimming, triathlon and gymnastics.
The fourth subsample consisted of 526 Danish recreational athletes recruited through social media channels between march and May 2020. The most common sports were: running (recreational), fitness (power and endurance), track and field, handball, triathlon, cycling, gymnastics and boxing.
- Measures: It is of interest why you did not include a control group of non-eating disordered and non-sports-active people in you analysis.
Response 3: We agree that it would have been interesting to include a sample of non-eating disordered and non-sports active individuals. However, our main aim was to test the factor structure for Danish use and as the EDE-Q is mainly used in clinical samples (and for our research also sports active individuals) we didn't have the premise to gather data from healthy and non-sports active individual. Also, if the factor structure is not solid in the main populations (persons with eating disorders), it is less relevant to test the factor structure in non-clinical samples.
- Measures: Further, why did you not include various cut-off points (e.g. >3 as a general cutoff or the male-specific of > 1.68 for males) regarding your point of construct validity?
Response 4: We did not operate with an EDE-Q cut-off point because we only examined the correlations between total scores of the different scales. Different cut-of points have been suggested for the EDE-Q during the last 10-15 years, but intentionally the scale was not developed to screen for potential eating disorder in large samples. It was developed as a clinical tool.
- Discussion, line 254: You state in this section that you psychometric evaluation did not find evidence for the original four-factor structure of the EDE-Q. It would be interesting what you suggest as consequences of this finding regarding the construction of the EDEQ as an instrument that is mostly used in eating disorder research.
Response 5: We have added some reflections to elaborate on our findings:
Our literature review further depicted the variance regarding number of factors (one, two or three) and items suggested (from 7 to 22) in previous validation studies. Thus, the internal structure of EDE-Q may be understood as a theoretical model useful in clinical settings, but it is not supported by empirical data.
Reviewer 2 Report
The manuscript has two different focuses: the review of the literature about EDE-Q scale factors in athletes and the validation of the Danish version of the EDE-Q scale. I think the paper could be interesting for the population included and for the clinical application that the scale. However, I have some concerns about the manuscript and about the methodology applied in the statistics:
- I think the review should be more discussed in the discussion. The authors reported a very interesting table but I think they synthesized too much the discussion about it.
- Have the authors evaluated the use of an explorative approach (Exploratory factor analysis) for the evaluation of the scale? The literature suggests that language translation could modify the factorial structures of the scale (as discussed by the authors), and for this reason, EFA should be applied. Especially because other translations found different factorial structure.
- Have you considered the effect of the COVID pandemia could have on the psychopathological scores as suggested by Monteleone et al., 2021/A (https://doi.org/10.1016/j.jad.2021.02.037) and Monteleone et al. 2021/B (https://doi.org/10.1007/s40519-020-01097-x)? Because this could have an effect on your data
Author Response
We would like to thank the editor and the reviewers for taking our manuscript into consideration. We are very thankful for all relevant comments, and we have responded to each of them. Thanks again for the collaboration about this article.
I think the paper could be interesting for the population included and for the clinical application that the scale. However, I have some concerns about the manuscript and about the methodology applied in the statistics:
I think the review should be more discussed in the discussion. The authors reported a very interesting table but I think they synthesized too much the discussion about it.
Response 1: We have expanded the discussion about the findings presented at the table. The results from the EDE-Q literature across several countries (table 1) indicate that no evident and consistent factor structure of the original EDE-Q can be identified. Models suggesting one-factor, two-factor and three-factor structures have been presented and evaluated. Shorter versions of EDE-Q have been developed, but the internal structure of the scale differ across studies and samples. Several studies point at a seven-item scale, but also 8-items, 15 items (for female), 16-items (for male), 18-items and 22-items have been proposed during the last 10 years.
- Have the authors evaluated the use of an explorative approach (Exploratory factor analysis) for the evaluation of the scale? The literature suggests that language translation could modify the factorial structures of the scale (as discussed by the authors), and for this reason, EFA should be applied. Especially because other translations found different factorial structure.
Response 2: We agree with the reviewer that an EFA could shed more light on particular factor structures in our four samples. Existing validation studies use CFA (Barnes et al. 2012; Grilo et al. 2014; Carrard et al. 2015; Kliem et al. 2016 etc.), while some studies do both CFA and EFA (Darcy et al. 2013; Forsén Manilla et al. 2017 etc).
Several studies using EFAs to assess the structure in their samples fail in formally validating the four-factor structure. The resulting (EFA) factor structures were highly inconsistent across studies and settings and could not subsequently be validated by other studies. This may suggest that exploratory factor structures of the EDE-Q are rather sample-specific and not generalizable. As our study comprises four different samples, it is likely that some of the EFAs for these four samples would result in different factor structures. Rather than adding (potentially four) further proposals to a factor structure of the EDE-Q, which would need subsequent validation, we have chosen to focus on the applicability of the EDE-Q in clinical practice due to the good construct validity of the scale and its use as tool for identifying eating disorders. We acknowledge the lack of evidence for the original four-factor structure and propose in our conclusion that future work may be done on developing a sport specific tool with higher reliability and validity.
We have mentioned the above points in the results section and added more information about the instability in factor structure:
The studies are presented in Table 1 and show remarkable inconsistency across samples. There is no evident factor structure, and the studies display both one-, two- and three-factor models” and “Different shorter versions (e.g., seven-items and eight-items) have been created and suggested, but none of them have been replicated sufficiently to conclude and recommend new evident factor solutions”.
And in the discussion section: “Further studies of the factor structure may confuse the landscape of psychometric reliability. Instead, we should evaluate the validity of EDE-Q because the instrument may be useful in identifying eating disorders” and “An exploratory analysis or another CFA model could have accounted for the inability of our CFA to provide an optimal fit of the original four-factor structure. However, the aim of our study was only to validate the existing gold standard scale in Danish and not to explore if another structure could be supported”.
And in the conclusion section: “Our literature review further depicted the variance regarding number of factors (one, two or three) and items suggested (from 7 to 22) in previous validation studies. Thus, the internal structure of EDE-Q may be understood as a theoretical model useful in clinical settings, but it is not supported by empirical data.
However, the construct validity was high in all samples, indicating that the scale has the potential to identify individuals with eating pathology and comorbid mental distress. Development of a sport specific tool may present with higher reliability and validity. Until then the EDE-Q is a usable screening tool, but with caution and awareness of its limitations.”
- Have you considered the effect of the COVID pandemia could have on the psychopathological scores as suggested by Monteleone et al., 2021/A (https://doi.org/10.1016/j.jad.2021.02.037) and Monteleone et al. 2021/B (https://doi.org/10.1007/s40519-020-01097-x)? Because this could have an effect on your data.
Response 3: Thanks for pointing out that the COVID-19 pandemia should be mentioned in our article.
A large part of our data was collected before the Covid-19 pandemia: 60% of the BED sample and 80-90% of the elite athlete sample. However, the clinical eating disorder sample was collected at the beginning of the crisis before the lockdown.
The results on construct validity would not be affected because the EDE-Q, SCOFF and BED-Q was completed at the same time.
The factor analysis might be affected because patients might rate some ED-symptoms higher than before the pandemia.
However, the discussion is speculative. The study by Monteleone et al. 2021 (https://doi.org/10.1016/j.jad.2021.02.037) used EDI-2 that consists of other sub-scales than EDE-Q. Furthermore, the study was retrospective and might be limited by recalling bias.
The study by Monteleone et al. 2021 was conducted in Italy, that was more severely affected by the pandemia compared to Denmark. The isolation period was longer, the number of Covid-related death was higher and the restrictions of society were more severe (wearing masks, not allowed to leave home, empty streets etc.).
We have added one article by Monteleone et al. 2021 in our discussion:
"It should be mentioned as a potential bias that some of our data were collected during the Covid-19 pandemic. A retrospective study from Italy [46] found that ED symptoms were increased in ED patients during March and April 2020. However, the crisis was worse in Italy than in Denmark including more restrictions, and more impact on hospitals, public health, and mortality."
Round 2
Reviewer 2 Report
I appreciate the authors' responses. I think they have clarified the points I raised. I think the paper could be considered for publication.